# Scanning Tunneling Microscopy (STM) Image Segmentation Using Unsupervised and Few-shot Learning

**Nikola Kolev[1,5], Emily Hoffman[1,5], Geoff Thornton[1,5], Max Trouton[1,5], Filippo Federici[2,3], David Gao[1,3,4], Steven Schofield[1,5], Taylor Stock[1,5], Neil Curson[1,5]**

(1) University College London, (2) Aalto University, (3) Nanolayers Research Computing LTD, (4) Norwegian Institute of Science and Technology, (5) London Centre for Nanotechnology

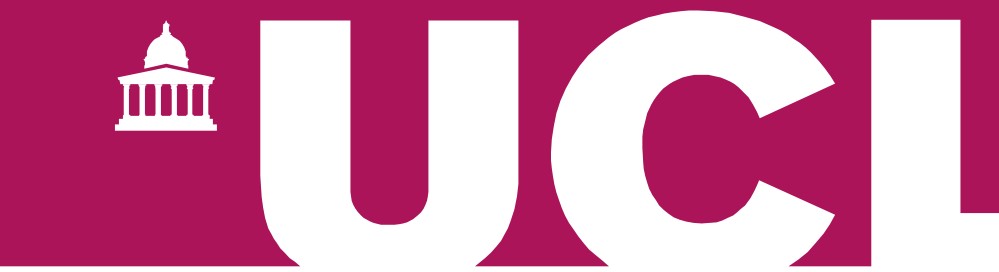

- Scanning tunneling microscopy (STM) is a powerful technique for imaging surfaces with atomic resolution, providing invaluable insights into surface structure and physical and chemical processes occurring on surfaces. A regular task of STM image analysis is detecting and labelling features of interest against the background of the unperturbed surface. Performing this segmentation manually is a labor-intensive task, requiring significant human effort.

- We propose an automated approach to the segmentation of STM images that leverages few-shot learning and unsupervised learning to remove the requirement for large manually annotated lattice datasets.

## (1) STM imaging

- A microscopy technique used to image conducting surfaces with **atomic resolution**. It can also **manipulate single atoms** on the surface.

- Each pixel in an image represents the height of the density of electron states at that point. We can measure either the filled or empty states, producing 2 channels.

Images of the Si(001) surface with different defects (the size of a few atoms) highlighted.

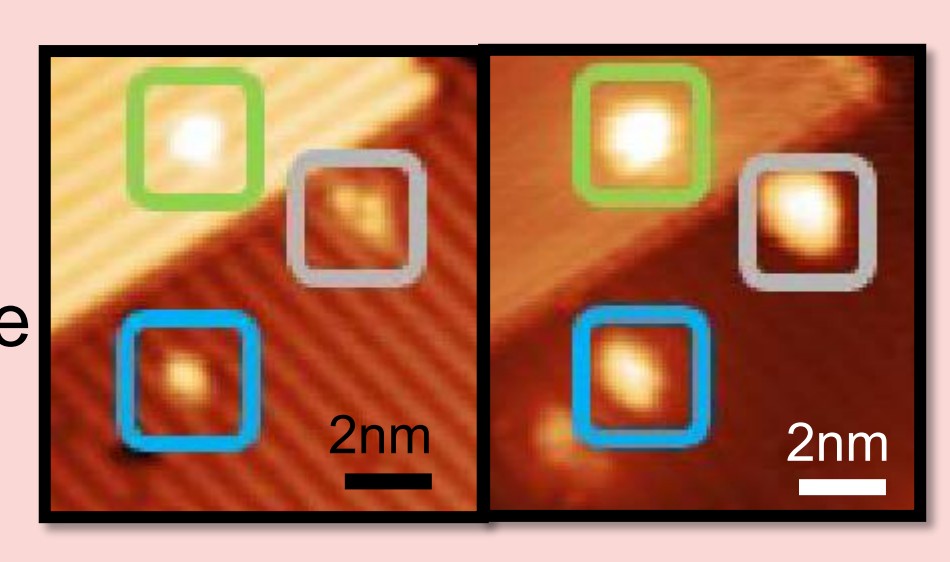

## (2) – UNet & Automated Labelling of training data

- UNet is used to produce a **binary map** of the all the defects on the surface (like the one shown in f).

- We want to **reduce** the time spent **manually** labelling training data for the UNet:

  - Use a pretrained network (FCNResnet101) to extract feature vectors for each pixel.

  - These are then clustered using k-means clustering to produce a segmented image.

  - By **varying the resolution** of the input, we can change how **detailed** the segmentation is: higher resolution highlights features such as **atomic rows and defects**, lower resolution focuses more on **phase domains.**

- These images are then augmented, and **extra experimental noise** is added to train a UNet. In this way, we get a more **robust**, and **faster**, segmentation network.

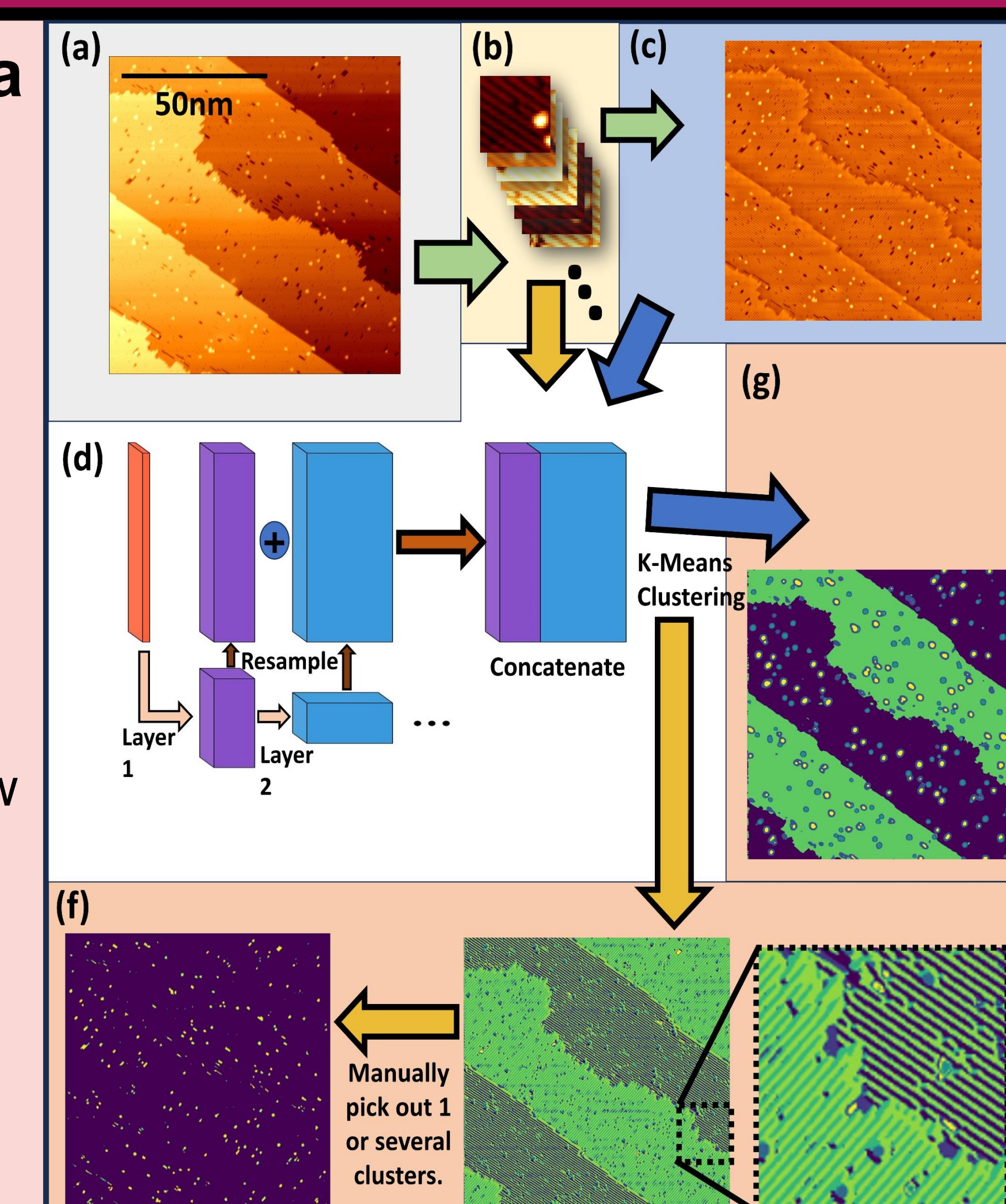

## (3) – FSL networks

- We test multiple few-shot learning (FSL) networks.

- The prototypical, matching, relation, and simple shot (conv4) all have a **Conv4 backbone** and are trained using episodes on **subject specific data**.

- We test a simple shot network with a pretrained (**non-subject specific**) Resnet18 backbone.

- Are the embeddings **useful/meaningful**? - We compare to the accuracy of KNN on the **bare** pixels.

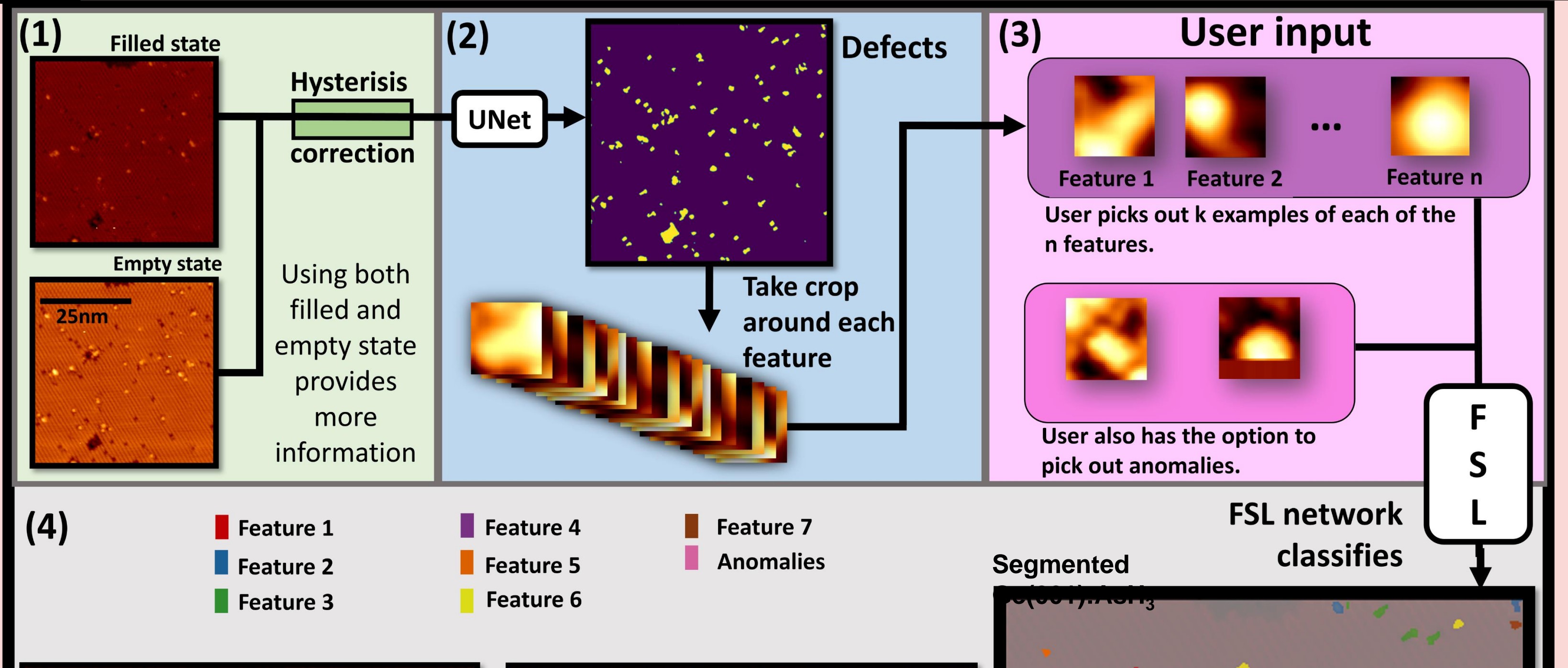

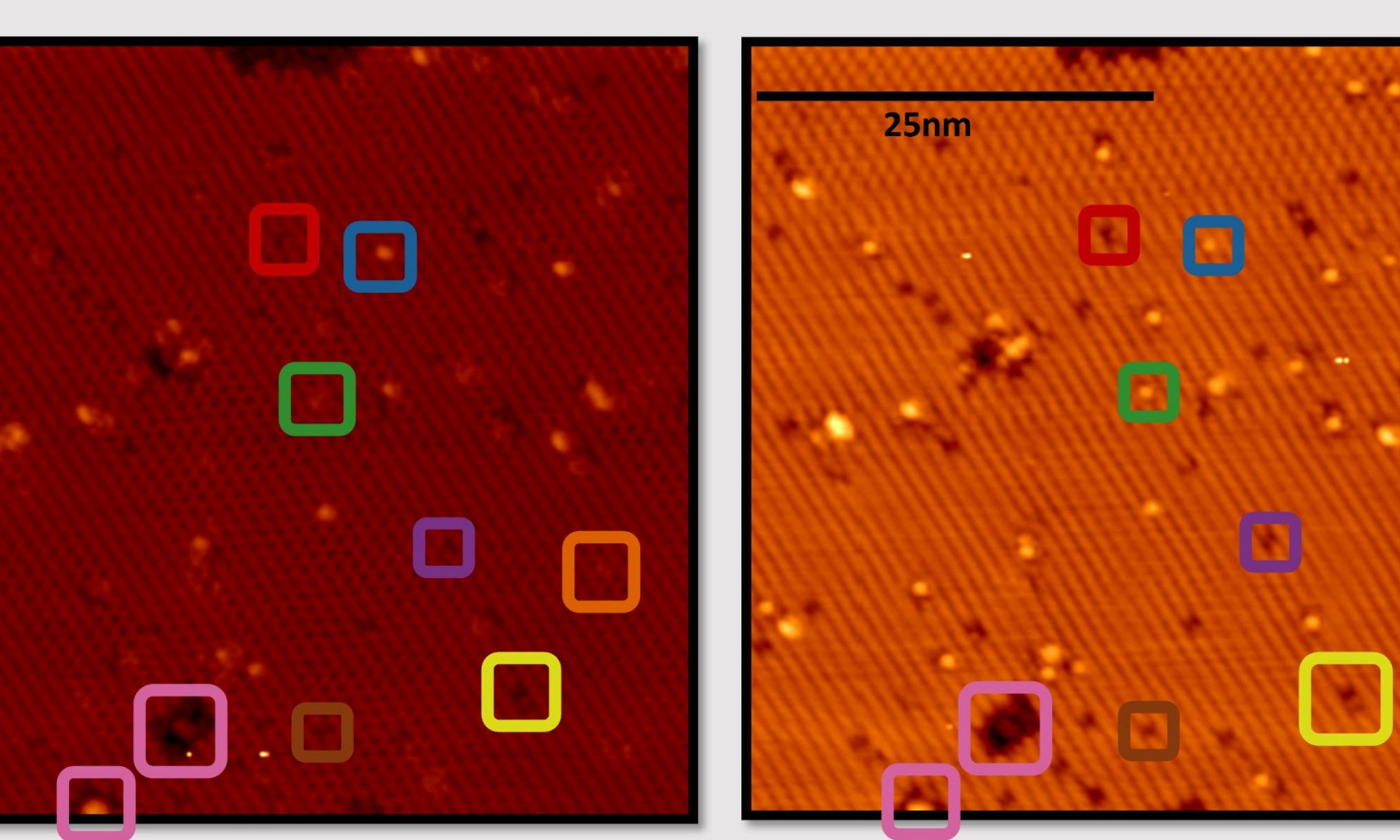

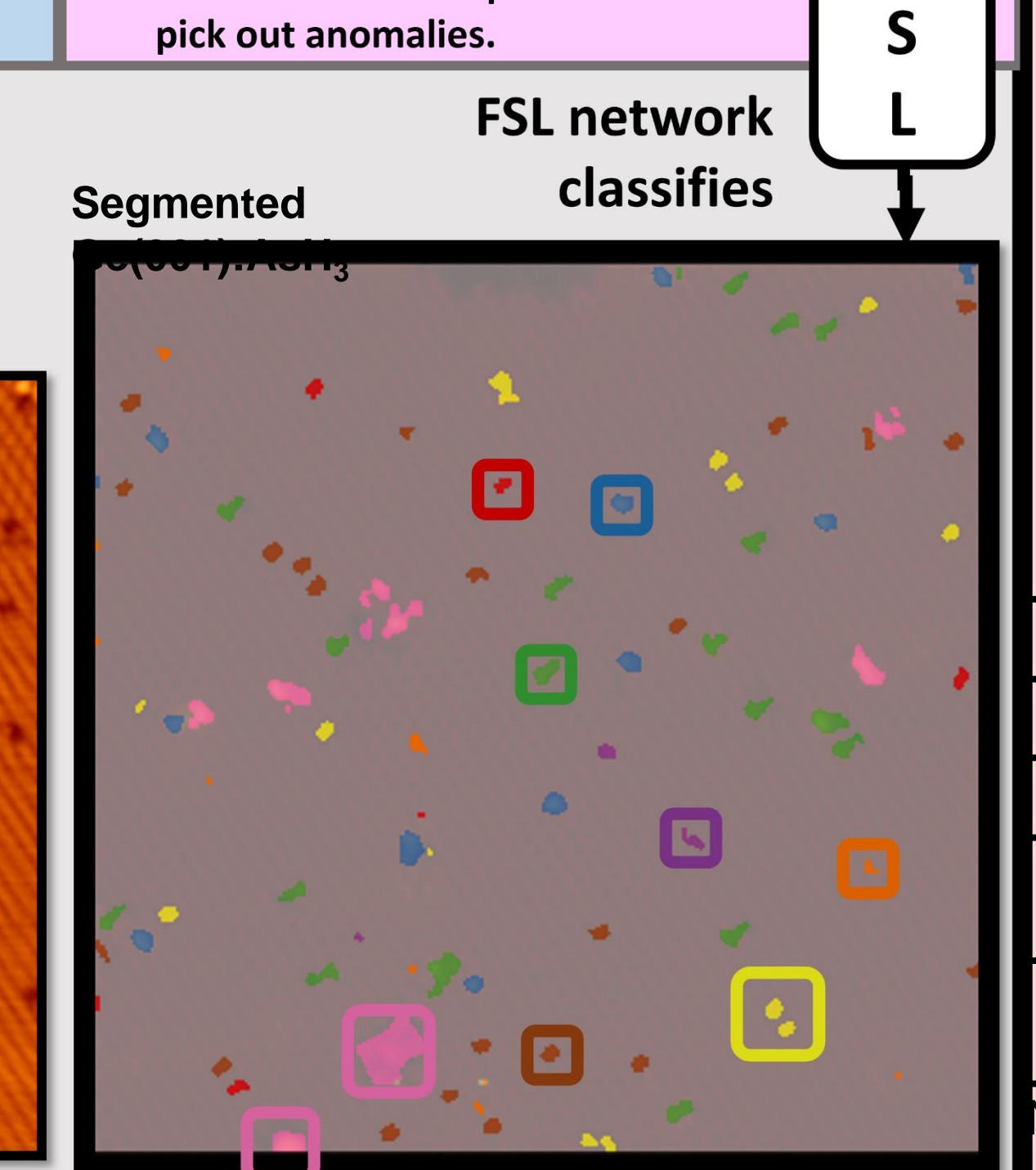

## (4i) – Si(001):H:AsH₃

- Surface is of especial significance for the **semiconductor** and **quantum computing** industry [1][2].

- FSL allows for flexibility to implement new dopant atom precursor types with as little as one new labelled data point.

- Models are trained and tested on data from the same surface.

| Model | Training Set | Acc (4-way, 1-shot) |
|---|---|---|
| Prototypical | Si defects | **95.567±0.013%** |
| Matching | Si defects | 94.950±0.009% |
| Relation | Si defects | 93.400%±0.014% |
| Simple shot (conv4) | Si defects | 92.933±0.010% |
| Simple shot (Resnet18) | ImageNet | 66.873±0.030% |
| NN (K=1) on bare pixels | Si defects | 76.567±0.020% |

## (4ii) – Ge(001):AsH₃ and TiO₂

Results for all tables are accuracies averaged over 100 episodes and with 95% confidence interval.

| Model | Training Set | Acc (4-way, 1-shot) |
|---|---|---|
| Prototypical | Si defects | 61.25±0.02% |
| Matching | Si defects | **61.61±0.02%** |
| Relation | Si defects | 25.07%±0.01% |
| Simple shot (conv4) | Si defects | 48.18±0.01% |
| Simple shot (Resnet18) | ImageNet | 43.00±0.02% |
| KNN (K=1) on bare pixels | Ge defects | 46.64.±0.02% |

Classification on Ge(001):AsH₃ data. **Trained on defects from non-Ge(001):AsH₃ data.**

| Model | Training Set | Acc (2-way, 1-shot) |
|---|---|---|
| Prototypical | Si & Ge defects | **70.03±0.03%** |
| Matching | Si defects | 61.60±0.02% |
| Relation | Si defects | 30.93%±0.03% |
| Simple shot (conv4) | Si defects | 54.47±0.02% |
| Simple shot (Resnet18) | ImageNet | 65.03±0.02% |
| KNN (K=1) on bare pixels | TiO₂ defects | 57.40±0.03% |

Classification on TiO₂(110) data. TiO₂(110) data has only filled state images. **Trained on defects from non-TiO₂(110) data.**

- The technique offers greater flexibility compared to previous supervised methods, being easier to adapt to an unseen surface while maintaining high accuracy, reaching up to 90%. This will make it useful for research which is constantly studying new substrates and adsorbates.

- Right hand column of tables shows accuracy of classification of the networks. It demonstrates the effectiveness of our approach on three distinct surfaces: Si(001):H:AsH₃, Ge(001):AsH₃, and TiO₂(110). We show that our model exhibits strong generalization capabilities, adapting well to unseen surfaces with only as little as one additional labeled data point after initial training.

- Different FSL-networks are tested, with the prototypical performing the best overall. The relation network shows signs of overfitting.

- Currently, no standardized dataset to use for benchmarking exists within the STM community. We believe this would be a worth while, but time consuming, venture.

- An ablation study (not included) showed simple manipulations to the data to generate new classes allowed for a better feature embedding and therefore accuracy.

[1] Stock, T.J., Warschkow, O., Constantinou, P.C., Li, J., Fearn, S., Crane, E., Hofmann, E.V., Kölker, A., McKenzie, D.R., Schofield, S.R. and Curson, N.J., 2020. Atomic-scale patterning of arsenic in silicon by scanning tunneling microscopy. ACS nano, 14(3), pp.3316-3327.

[2] Stock, T.J., Warschkow, O., Constantinou, P.C., Bowler, D.R., Schofield, S.R. and Curson, N.J., 2024. Single-Atom Control of Arsenic Incorporation in Silicon for High-Yield Artificial Lattice Fabrication. Advanced Materials, p.2312282.

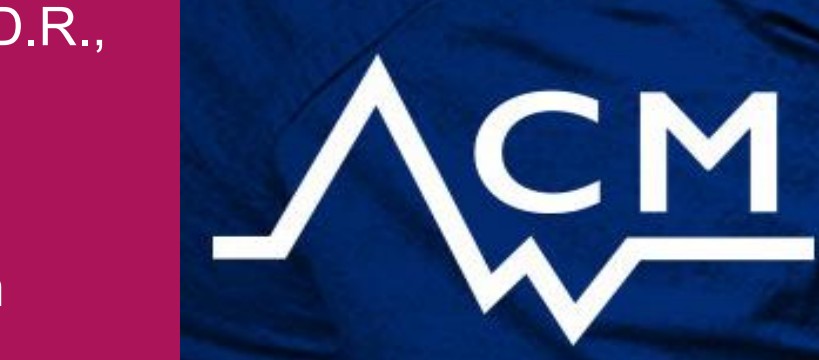
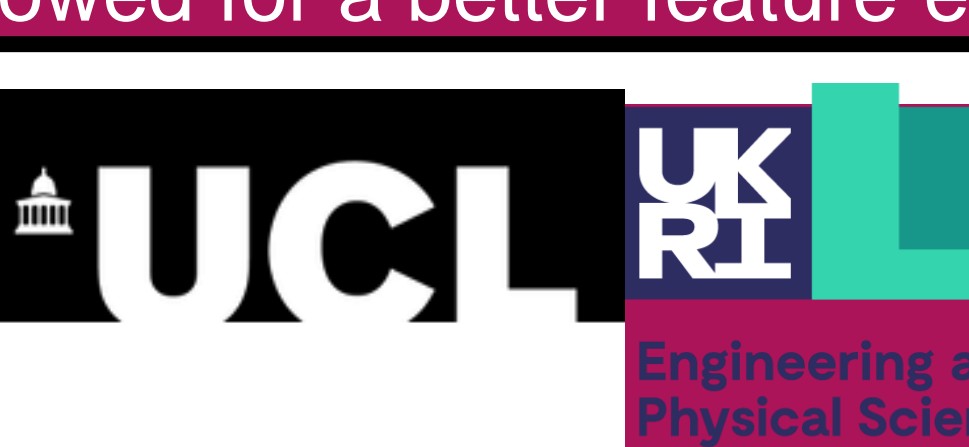
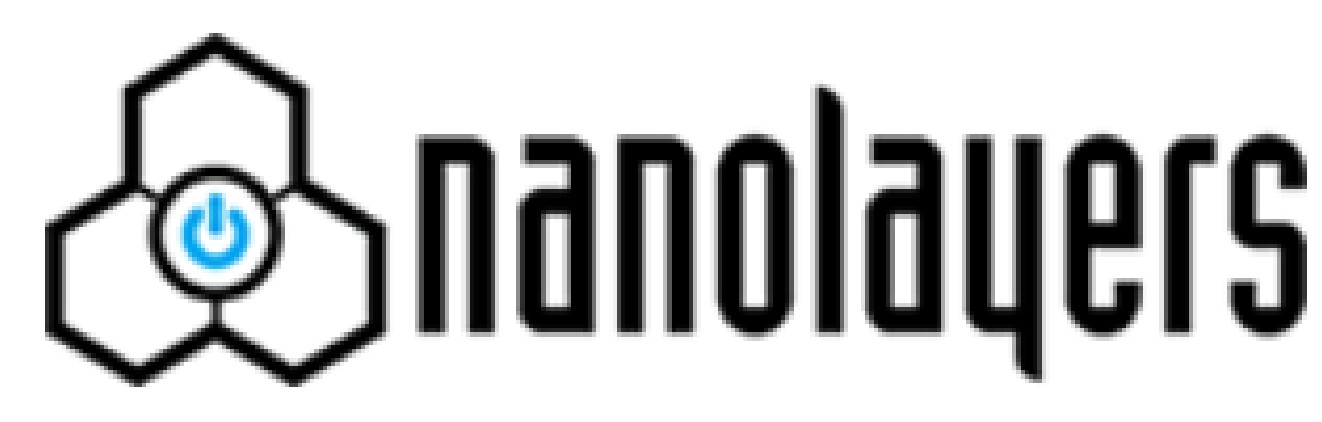