# OpenReview forum: "Scanning Tunneling Microscopy (STM) Image Segmentation Using Unsupervised and Few-shot Learning"
_ICML.cc/2024/Workshop/ML4LMS — ML4LMS Poster_

### Official Review · Reviewer_MnjA · 2024-06-11
**Automated labeling of training data**

**Rating:** 7
**Confidence:** 1

**Review:**

-

---

### Official Review · Reviewer_Rrda · 2024-06-12
**STM Image Segmentation**

**Rating:** 4
**Confidence:** 4

**Review:**

The models used are not state of the art and also there is huge potential for finetuning models like SAM and then try on such tasks.

---

### Official Review · Reviewer_sbDY · 2024-06-12
**Review for poster: Scanning Tunneling Microscopy (STM) Image Segmentation Using Unsupervised and Few-shot Learning**

**Rating:** 7
**Confidence:** 4

**Review:**

In general the poster is good, manual labeling is greatly reduced.
In the meantime, we would like to see more benchmarks to see the improvement, and to understand why we have this improvement.
And the accuracies across dataset have large deviations across all models, would like to know more on data.